# Introducing CuCo_2_S_4_ Nanoparticles on Reduced Graphene Oxide for High-Performance Supercapacitor

**DOI:** 10.3390/nano14020182

**Published:** 2024-01-12

**Authors:** Xue Fang, Cong Yang, Xiaochen Zhang, Yang Wang, Jiali Yu

**Affiliations:** 1Institute of Advanced Technology, Heilongjiang Academy of Sciences, Harbin 150001, China; xiaoxue@iathas.ac.cn (X.F.); xiaochen@iathas.ac.cn (X.Z.); wangyang@iathas.ac.cn (Y.W.); 2Institute of Low-dimensional Materials Genome Initiative, College of Chemistry and Environmental Engineering, Shenzhen University, Shenzhen 518060, China; yang@iathas.ac.cn

**Keywords:** graphene oxides, metallic sulfides, CuCo_2_S_4_ nanoparticles, supercapacitor, composites

## Abstract

In this work, a bimetallic sulfide-coupled graphene hybrid was designed and constructed for capacitive energy storage. The hybrid structure involved decorating copper–cobalt–sulfide (CuCo_2_S_4_) nanoparticles onto graphene layers, with the nanoparticles anchored within the graphene layers, forming a hybrid energy storage system. In this hybrid structure, rGO can work as the substrate and current collector to support the uniform distribution of the nanoparticles and provides efficient transportation of electrons into and out of the electrode. In the meantime, CuCo_2_S_4_-active materials are expected to offer an evident enhancement in electrochemical activities, due to the rich valence change provided by Cu and Co. Benefiting from the integrated structure of CuCo_2_S_4_ nanoparticles and highly conductive graphene substrates, the prepared CuCo_2_S_4_@rGO electrode exhibited a favorable capacitive performance in 1 M KOH. At 1 A g^−1^, CuCo_2_S_4_@rGO achieved a specific capacitance of 410 F g^−1^. The capacitance retention at 8 A g^−1^ was 70% of that observed at 1 A g^−1^, affirming the material’s excellent rate capability. At the current density of 5 A g^−1^, the electrode underwent 10,000 charge–discharge cycles, retaining 98% of its initial capacity, which indicates minimal capacity decay and showcasing excellent cycling performance.

## 1. Introduction

Over recent decades, supercapacitors (SCs) have garnered significant attention as a dependable electrochemical energy storage (EES) system due to their extended lifespan and high power density. The advent of pseudocapacitors, originally introduced by Conway, has spurred extensive investigation owing to their rapid and reversible faradic reactions occurring at the surface [1,2,3]. Lately, scientists have focused on exploring the synthesis of bimetallic compounds using various transition metal elements such as Co, Ni, Fe, Mn, Mo, and so on for supercapacitor applications. Particularly, Manickam Minakshi et al. fabricated Nano α-NiMoO_4_ and CoMoO_4_ as new active materials for electrochemical supercapacitors, which demonstrated excellent energy storage performances [4,5,6]. In addition to binary metal oxides, binary metal sulfides have shown remarkable properties such as enhanced electrical conductivity, increased redox sites, and expansive oxidation states resulting from the synergistic combination of two metal sulfides within a single molecule [7,8]. To date, several binary transition metal sulfides, including NiCo_2_S_4_, MnCo_2_S_4_, CuCo_2_S_4_, ZnCo_2_S_4_, and FeCo_2_S_4,_ have found successful applications in the realm of SCs [9,10,11]. Notably, compared to other transition metal compounds, CuCo_2_S_4_, initially synthesized by Ye et al., is considered to be one of the most promising electrode materials due to its environmental friendliness and the low cost of Cu. By replacing the oxygen atom in CuCo_2_O_4_ with sulfur atoms, CuCo_2_S_4_ can be obtained with the well-preserved spinel structure of the original compound. CuCo_2_S_4_ exhibits enhanced electron transport due to its higher electronegativity and smaller bandgap as compared with CuCo_2_O_4_. This superior electron transport capability is believed to positively impact on the electrochemical performance. As a result, CuCo_2_S_4_ nanoparticles synthesized by Chen et al. through a hydrothermal process exhibited a specific capacity of 90.6 mAh g^−1^ at a current density of 2 A g^−1^. Impressively, these nanoparticles retained 47.9% of their capacity when the current density was increased from 2 to 50 A g^−1^. However, challenges persist with CuCo_2_S_4_ electrodes, primarily stemming from issues such as self-agglomeration and poor conductivity, leading to limitations in the rate capability and cycle stability [12,13,14].

To address this challenge, the strategic amalgamation of CuCo_2_S_4_ with a highly conductive carbon matrix becomes imperative. Fortunately, graphene emerges as an optimal candidate owing to its exceptional conductivity and expansive surface area. When graphene is combined with inorganic materials such as metal oxides/hydroxides/sulfides, unique structural features and synergistic electrochemical properties can be imparted to the resulting graphene-based nanohybrids [15,16,17,18]. In fact, various functional nanohybrids comprising metal oxides or hydroxides have been successfully constructed to achieve a higher pseudocapacitance and enhanced electrochemical performance. Moreover, based on prior findings, the penetration depth of electrolyte ions into the electrode material is approximately 20 nm. Hence, reasonably reducing the size of active materials is a desired approach [19,20,21].

Inspired by prior works, we fabricated the CuCo_2_S_4_@rGO composite electrode using a straightforward two-step solvothermal approach in this study. The incorporation of CuCo_2_S_4_ nanoparticles is expected to offer an evident enhancement in electrochemical activities, due to the rich valence change provided by Cu and Co. Simultaneously, reduced graphene oxide (rGO) plays a multifaceted role: providing nucleation sites for CuCo_2_S_4_ growth, preventing the agglomeration of CuCo_2_S_4_ nanoparticles, and serving as a conductive scaffold, thereby amplifying the conductivity of CuCo_2_S_4_@rGO. Furthermore, by depositing the CuCo_2_S_4_/rGO composite onto nickel foam, the resultant product can function as a freestanding supercapacitor electrode for energy storage purposes. Benefiting from the integrated structure of CuCo_2_S_4_ nanoparticles and highly conductive graphene substrates, the prepared CuCo_2_S_4_ electrode exhibited a favorable capacitive performance in 1 M KOH. At 1 A g^−1^, CuCo_2_S_4_@rGO achieved a specific capacitance of 410 F g^−1^. The capacitance retention at 8 A g^−1^ was 70% of that observed at 1 A g^−1^, affirming the material’s excellent rate capability. At the current density of 5 A g^−1^, the electrode underwent 10,000 charge–discharge cycles, retaining 98% of its initial capacity, which indicates minimal capacity decay and showcasing excellent cycling performance.

## 2. Experimental Section

### 2.1. Preparation of CuCo_2_O_4_@rGO

An amount of 15 mg of GO powder (Zhongke, Shanghai, China) was dispersed in 5 mL of deionized (DI) water, forming a 3 mg mL^−1^ GO aqueous dispersion. Then, 15 mL of a 1:1 mixture of DI water and ethanol was added into the GO dispersion, labeled as solution A. Amounts of 483.2 mg of Cu(NO_3_)_2_ and 582.1 mg of Co(NO_3_)_2_ were dispersed in 10 mL of DI water, labeled as solution B.

Solution A was slowly dripped into solution B while stirring continuously for 2 h. Then, the resulting mixture was transferred into a polytetrafluoroethylene reaction vessel and a hydrothermal reaction was conducted at 160 °C for 6 h. After the reaction, the obtained black sample was filtered, washed, and dried. Finally, the sample was annealed at 500 °C for 1 h in a tube furnace to enhance the crystallinity of the sample. This product is denoted as CuCo_2_O_4_@rGO.

### 2.2. Preparation of CuCo_2_S_4_@rGO

Amounts of 25 mg of the aforementioned CuCo_2_S_4_@rGO sample, along with 100 mg of TAA, were added into 30 mL of anhydrous ethanol to form a homogeneous dispersion. This mixture was then transferred into a polytetrafluoroethylene reaction vessel and a hydrothermal reaction was conducted at 180 °C for 6 h.

After the reaction, the resulting product was washed 5 times with DI water and then vacuum-dried to obtain the CuCo_2_S_4_@rGOsample.

### 2.3. Preparation of CuCo_2_S_4_@rGO@nickel Foam Freestanding Electrode

A commercial nickel foam material was selected as the electrode substrate, and a slurry of CuCo_2_S_4_@rGO material was prepared following a specific method. The slurry was uniformly applied onto the nickel foam substrate using a blade-coating method, creating a highly flexible nickel foam-based electrode. The slurry preparation method is outlined as follows:

The nickel foam was cut into rectangular pieces measuring 1 cm × 3 cm, then subjected to ultra-sonication in water and ethanol for 30 min, respectively, to remove the surface impurities, followed by overnight vacuum-drying.

A specific amount of polyvinylidene fluoride (PVDF) was put into a mortar with a small amount of NMP (N-methyl-2-pyrrolidone), and ground until the PVDF was completely dissolved. Then, a specific amount of conductive carbon black (acetylene black) and CuCo_2_S_4_@rGO was added into the PVDF, continuing to grind until a uniform, relatively viscous black slurry was formed.

Due to the hygroscopic nature of PVDF, all materials and tools need to be moisture-free. Pre-dissolving PVDF in an NMP solution prevents it from coagulating due to air humidity. The mass ratio of the slurry comprises PVDF (binder), conductive agent (acetylene black), and electrode-active material in a ratio of 1:1:8. The addition of NMP is empirical, typically within a range of approximately 5 to 10 drops, which may be adjusted according to the specific materials used.

Finally, the slurry is blade-coated onto the front 1 cm × 1 cm area of the foam nickel. An average coating is achieved by using a syringe needle to evenly distribute the slurry. The electrode is then placed in a vacuum oven and dried at 80 °C overnight.

### 2.4. Measurements

The phase constituents and crystal structure of the specimens were analyzed using powder X-ray diffractometry (XRD) conducted on a MiniFlex600 instrument from Rigaku, Tokyo, Japan. The measurements were performed at 40 kV and 15 mA with nickel-filtered Cu-Kα radiation over an angular range spanning from 3° to 80°, at a scanning rate of 5° per minute. To characterize the morphology and structure of the samples, field-emission scanning electron microscopy (FESEM) was conducted using a JSM-7800F instrument (Tokyo, Japan), while transmission electron microscopy (TEM) analysis was carried out using a JEM2100 instrument (Tokyo, Japan). Additionally, X-ray photoelectron spectroscopy (XPS) measurements were conducted utilizing an Escalab-250Xi electron spectrometer manufactured by Thermo Scientific (Waltham, MA, USA), with Al-Kα radiation employed as the X-ray source.

The electrochemical characteristics of CuCo_2_S_4_@rGO, employed as the working electrode, were examined within a three-electrode cell containing a 3 M KOH aqueous solution. Platinum plate and Hg/HgO were utilized as the counter and reference electrodes, respectively, while the mass of CuCo_2_S_4_@rGO stood at 2.0 mg. Various electrochemical techniques including cyclic voltammetry (CV), galvanostatic charge–discharge (GCD), and electrochemical impedance spectroscopy (EIS) were performed to assess its electrochemical performance using an electrochemical workstation (Yicheng Hengda Technology Co., Ltd., Beijing, China). Deriving from the galvanostatic discharge curve, the specific capacitance (*Cs*) was calculated utilizing the following equation:(1)$$Cs=\frac{I∆t}{m∆V}$$

Here, *I* (A) represents the steady discharge current, while *m* (g) stands for the weight of the active material within the electrode. ∆*t* (s) denotes the constant discharge duration, and ∆*V* (V) signifies the potential window utilized [22].

## 3. Results and Discussion

The CuCo_2_O_4_@rGO sample, obtained after the initial annealing step, was examined using field-emission scanning electron microscopy (SEM), as depicted in Figure 1a,b. The observations revealed a morphology of plenty of nanoparticles encapsulated within the graphene layers. From the subsequent transmission electron microscopy (TEM) investigation (Figure 1c–e), it can be seen that the distributions of the nanoparticles are very uniform across the whole thin graphene layers. The nanoparticles demonstrate an irregular flake morphology with a lateral size range of 10~30 nm.

The sulfidation process took place within a hydrothermal reactor, utilizing thioacetamide (TAA) as the sulfur source and anhydrous ethanol as the solvent. During the hydrothermal process, the GO is further reduced during the solvothermal process due to the reducibility of S^2−^ derived from TAA, resulting in enhanced electrical conductivity. Then, the copper cobalt oxides were decomposed as the temperature rose and transformed into CuCo_2_S_4_ nanoparticles, finally anchoring on the rGO. The reaction proceeded at a temperature of 180 °C for 6 h, resulting in the formation of uniformly distributed particulate CuCo_2_S_4_ on the graphene sheets, as depicted in Figure 2. The particles exhibited uniform diameters and were evenly dispersed within the graphene layers. Through TEM analysis, the sample morphologies were examined. As depicted in Figure 2c,d, nanoscale sulfide particles with diameters ranging from 5 to 10 nm can be observed, which are uniformly anchored within the layers of oxidized graphene. Graphene encapsulated the nanoparticles, displaying a homogeneously stable morphology. Since graphene possesses excellent electronic conductivity, the close contact between the nanoparticle and graphene could provide expedited pathways for electron transport and facilitate energy storage kinetics. The clearly visible lattice fringes, exhibiting lattice spacings of 0.55 nm and 0.33 nm, correspond to the (111) and (220) crystal planes of cubic CuCo_2_S_4_, respectively.

The X-ray diffraction (XRD) analysis of the samples revealed distinctive patterns. Figure 3a represents the XRD spectrum of CuCo_2_O_4_@rGO. By comparing the sample’s XRD peaks with the standard PDF cards, we identified the predominant presence of CuCo_2_O_4_, corresponding to the JCPDS 001-1155 [23] standard card.

After sulfuration, as shown in Figure 3b, the crystalline structure of CuCo_2_S_4_@rGO is identified by the XRD technique. The diffraction peaks can be assigned to the cubic phase of spinel CuCo_2_S_4_ based on JCPDS No.42−1450 [24], respectively. Moreover, the low peak intensity and other peaks that appeared between 15° and 90° indicate the poor crystallization and impurity of CuCo_2_S_4_ [25]. On the base of the XRD patterns, the crystalline structure model of CuCo_2_S_4_ is revealed in Figure 3b.

Figure 4 presents the Raman spectra of the CuCo_2_S_4_@rGO sample. For comparison, the Raman spectra of graphene oxide (GO) that was used as the precursor for preparing the rGO in the CuCo_2_S_4_@rGO sample is also provided. Notably, the characteristic peaks corresponding to defect (D) and graphite (G) carbons were observed in both of the two samples at approximately 1350 cm^−1^ and 1580 cm^−1^ wavelengths, respectively. The I_D_/I_G_ ratio of pristine GO is 0.89. While after the hydrothermal process, the I_D_/I_G_ ratio for the CuCo_2_S_4_@rGO sample decreased to 0.72, indicating an increment in graphitization in the material. This enhancement promotes better electron conduction within the graphene layers [26]. Interestingly, the Raman spectrum of the CuCo_2_S_4_@rGO sample did not exhibit distinct peaks attributable to the embedded metal sulfides. This absence may stem from either the relatively low content of metal sulfides or the low intensity of the diffraction peaks.

By applying X-ray photoelectron spectroscopy (XPS) to analyze the surface chemical information of the CuCo_2_S_4_@rGO sample (Figure 5), the broad scan spectrum indicates the presence of six primary elements, S, C, N, O, Co Cu, and Co, in the sample. The atomic ratios of the different elements are listed in Table 1.

By deconvoluting the fine spectra of Cu 2p, Co 2p, C 1s, and S 2p, the chemical composition and electronic bonding states of the elements were analyzed. The high-resolution XPS spectrum of Cu 2p (Figure 6a) demonstrates Cu 2p3/2 and Cu 2p1/2 peaks along with two satellite peaks (marked as “sat”). Cu 2p3/2 and Cu 2p1/2 can be further divided into two spin–orbit doublets, indicating the coexistence of Cu^+^ and Cu^2+^. The Cu^+^ peaks of Cu 2p1/2 and Cu 2p3/2 were identified at 952.6 eV and 932.75 eV, respectively, with a splitting energy of 19.85 eV. Additionally, two dominant peaks corresponding to Cu^2+^ were resolved at 935.94 eV and 955.92 eV. The Co 2p spectrum (Figure 6b) displayed two peaks at the binding energies of near 781 eV and 797 eV, corresponding to Co 2p3/2 and Co 2p1/2, respectively, which is consistent with the literature [27]. Co 2p3/2 and Co 2p1/2 can be further split into two spin–orbit lines, evidencing the presence of divalent and trivalent Co species. Figure 6c represents the C 1s fine spectrum, revealing deconvoluted peaks attributed to C–C, C–O, and C=O bonds [28]. The deconvoluted peaks of the S 2p spectrum exhibited two primary peaks centered at 161.8 eV and 163.1 eV, which are the characteristic peaks of metal sulfides, corresponding to sulfides and disulfides, respectively [29]. Additionally, a significant peak around 165 eV corresponding to C–S–C bonds can be observed, indicating the bonding between sulfides and graphene layers. The weak SOx peak near 169.5 eV represents a low sulfur content in the oxide, which is due to the shortcomings of the liquid-phase hydrothermal sulfidation method [30]. The XPS results confirm the successful construction of a well-defined binary metal sulfide stably bonded on the reduced graphene oxide layers. Since the binary metal sulfide is highly active for electrochemical reactions and the rGO is conductive, the combination of these two materials could provide a synergic effect for energy storage. At first, rGO can work as the substrate and the current collector supports the uniform distribution of the nanoparticles and provides the efficient transportation of electrons into and out of the electrode. Secondly, CuCo_2_S_4_-active materials are expected to offer an evident enhancement in electrochemical activities, due to the rich valence change provided by Cu and Co. As a result, a high specific capacitance and rate performance were achieved by the as-synthesized CuCo_2_S_4_@rGO.

The electrochemical performances of the CuCo_2_S_4_@rGO sample were evaluated in a three-electrode system with 1 M KOH aqueous solution as the electrolyte, Hg/HgO as the reference electrode, and Pt as the counter electrode. For the convenience of testing, CuCo_2_S_4_@rGO was loaded on a nickel foam as a freestanding electrode. Figure 7a shows the cyclic voltammetry (CV) curves of CuCo_2_S_4_@rGO at scan rates ranging from 5 to 100 mV s^−1^ within a potential window of 0~0.8 V. It can be seen from Figure 7a that a pair of broad redox peaks can be clearly observed in all the CV curves, indicating that the capacitances are mainly derived from the pseudocapacitance associated with the reversible faradic reactions of the metal sulfide [31,32]. Furthermore, a minor shift in the peak current position can be observed with the increasing scan rates, attributed to the polarization phenomenon. At the scan rate of 100 mV s^−1^, the CV maintained its shape observed at lower scan rates, verifying the superior kinetic reversibility and rate performance of the sample. The observed redox peaks are attributable to reversible faradaic reactions, as depicted by the following equations:(2)$$CuCo_{2}S_{4}+OH^{-}+H_{2}O↔CuS_{4-4X}OH+CoS_{2X}OH+e^{-}$$
(3)$$CoS_{2X}OH+OH^{-}↔CoS_{2X}O+H_{2}O+e^{-}$$
(4)$$CoS_{2X}+OH^{-}↔CoS_{2X}OH+e^{-}$$
(5)$$CuS_{4-4X}+OH^{-}↔CuS_{4-4X}OH+e^{-}$$

The GCD curves at the current densities ranging from 1 A g^−1^ to 8 A g^−1^ reveal symmetric shapes with almost equivalent charge–discharge durations, manifesting the good columbic efficiency of the sample (Figure 7b). Moreover, it should be noted that as the current density increases, no visible IR drops can be observed in the curves obtained at high current densities, indicating the excellent conductivity of CuCo_2_S_4_@rGO, which is meaningful for the high rate performance. In fact, from the electrode impedance spectroscopy (EIS) results of the sample, the *Rs* and *Rct* of the sample is only about 0.85 Ω and 1.55 Ω, respectively (Figure 7d). The calculated specific capacitance based on the GCD curves is presented in Figure 7c. At current densities of 1 A g^−1^, 2 A g^−1^, 3 A g^−1^, 4 A g^−1^, 5 A g^−1^, 6 A g^−1^, 7 A g^−1^, and 8 A g^−1^, the specific capacities were 415 F g^−1^, 376 F g^−1^, 355 F g^−1^, 338 F g^−1^, 325 F g^−1^, 313 F g^−1^, 303 F g^−1^, and 290 F g^−1^, respectively. The capacitance retention at 8 A g^−1^ was 70% of that observed at 1 A g^−1^, affirming the material’s excellent rate capability [33]. The results demonstrated above manifest that the conductivity and pseudocapacitive performance provided by the rGO substrate and uniformly distributed CuCo_2_S_4_ nanoparticles are favorable for the fast diffusion of electrolyte ion and electron transportations during electrochemical reactions, thus leading to a higher electrochemical activity and energy storage performance.

Figure 8 displays the cyclic performance of CuCo_2_S_4_@rGO in the three-electrode system, demonstrating exceptional cycling stability. At the current density of 5 A g^−1^, the electrode underwent 10,000 charge–discharge cycles, retaining 98% of its initial capacity, which indicates minimal capacity decay and showcasing excellent cycling performance.

Table 2 illustrates the comparison of the energy storage performance obtained in our work with other recently reported sulfide–graphene composite nanomaterials [34]. As can be observed, the specific capacitance reported in this work is comparable to or exceeds many of the existing counterpart works, although the comparative analysis indicates that the specific capacitance achieved in this study is superior. However, there is still room for optimization to further increase the energy storage of CuCo_2_S_4_@rGO. The optimization techniques include modifying the morphologies of CuCo_2_S_4_ and the loading densities of CuCo_2_S_4,_ which will be explored in our future work.

The excellent electrochemical performance observed in the CuCo_2_S_4_@rGO electrode can be attributed to several key factors. Firstly, reduced graphene oxides serve as a conductive matrix and scaffold for CuCo_2_S_4_ deposition. This role facilitates electron transport and prevents CuCo_2_S_4_ agglomeration, thus enhancing electrode material utilization and rate capability. Secondly, the presence of CuCo_2_S_4_ nanoparticles effectively reduces the distance for ion transmission and provides ample active sites for faradaic redox reactions, thereby amplifying the electrochemical performance. Thirdly, the direct growth of CuCo_2_S_4_ nanoparticles on rGO mitigates the stress induced during cycling, thereby bolstering the stability of CuCo_2_S_4_ within the electrode system.

## 4. Conclusions

In this work, we decorated rGO with pseudocapacitive active CuCo_2_S_4_ nanoparticles to address the issues of the poor cyclic performance and low rate capability observed in pseudocapacitive materials. Employing a hydrothermal synthesis method, CuCo_2_S_4_@rGO composites were successfully synthesized, followed by low-temperature annealing to enhance the crystallinity of the metal oxides. Subsequently, utilizing TAA as a sulfur source, a hydrothermal sulfidation process successfully yielded CuCo_2_S_4_@rGO nanocomposites. The results indicate the superior performance of the CuCo_2_S_4_@rGO sample. In a 1 M KOH electrolyte, at 1 A g^−1^, CuCo_2_S_4_@rGO achieved a specific capacitance of 410 F g^−1^. The capacitance retention at 8 A g^−1^ was 70% of that observed at 1 A g^−1^, affirming the material’s excellent rate capability. At the current density of 5 A g^−1^, the electrode underwent 10,000 charge–discharge cycles, retaining 98% of its initial capacity, which indicates minimal capacity decay and showcasing an excellent cycling performance.

## Figures and Tables

**Figure 1 nanomaterials-14-00182-f001:**
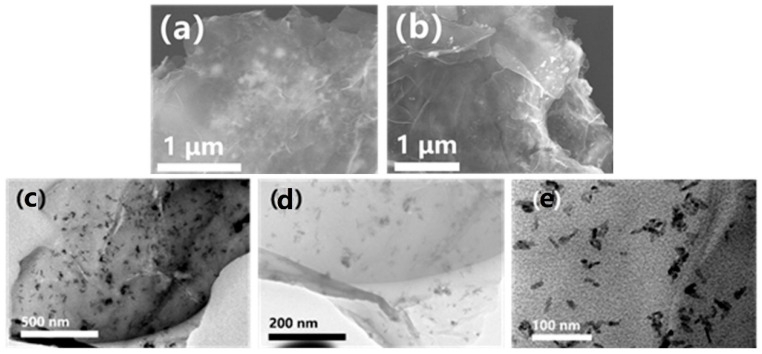
(**a**,**b**) FESEM images of CuCo_2_O_4_@rGO sample. (**c**–**e**) TEM images of CuCo_2_O_4_@rGO sample.

**Figure 2 nanomaterials-14-00182-f002:**
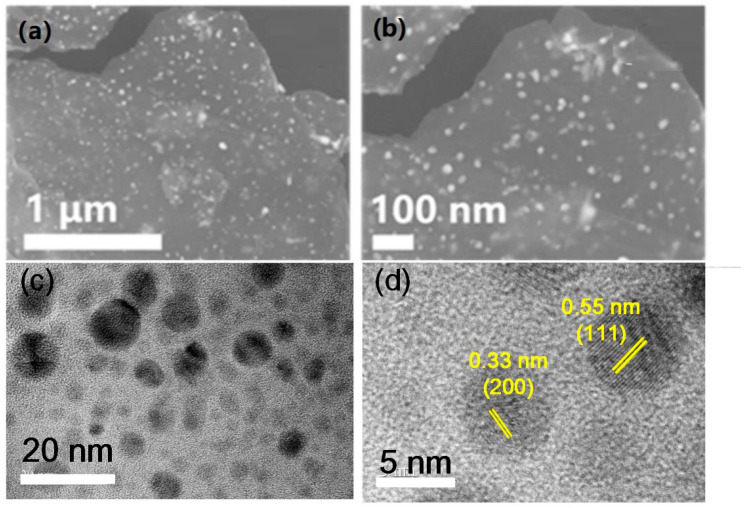
(**a**,**b**) SEM images of CuCo_2_S_4_@rGO sample; (**c**,**d**) TEM images of CuCo_2_S_4_@rGO sample.

**Figure 3 nanomaterials-14-00182-f003:**
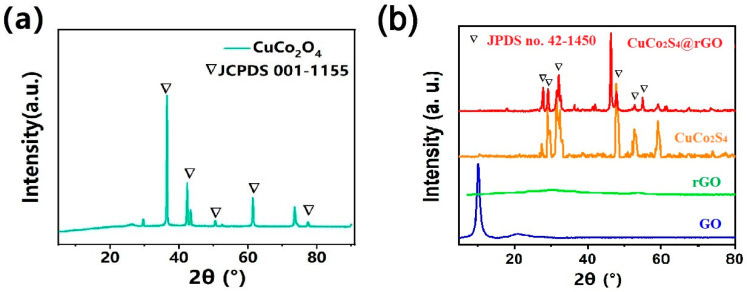
(**a**) XRD patterns of CuCo_2_O_4_@rGO; (**b**) XRD patterns of GO, rGO, CuCo_2_S_4,_ and CuCo_2_S_4_@rGO.

**Figure 4 nanomaterials-14-00182-f004:**
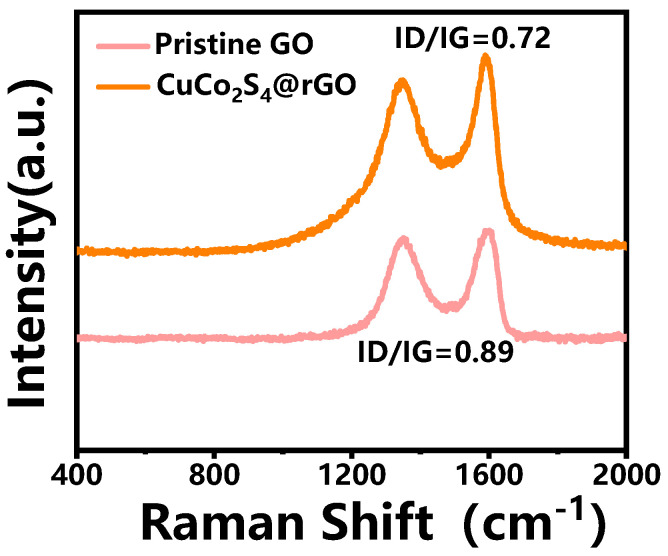
Raman spectra of CuCo_2_S_4_@rGO sample and pristine GO.

**Figure 5 nanomaterials-14-00182-f005:**
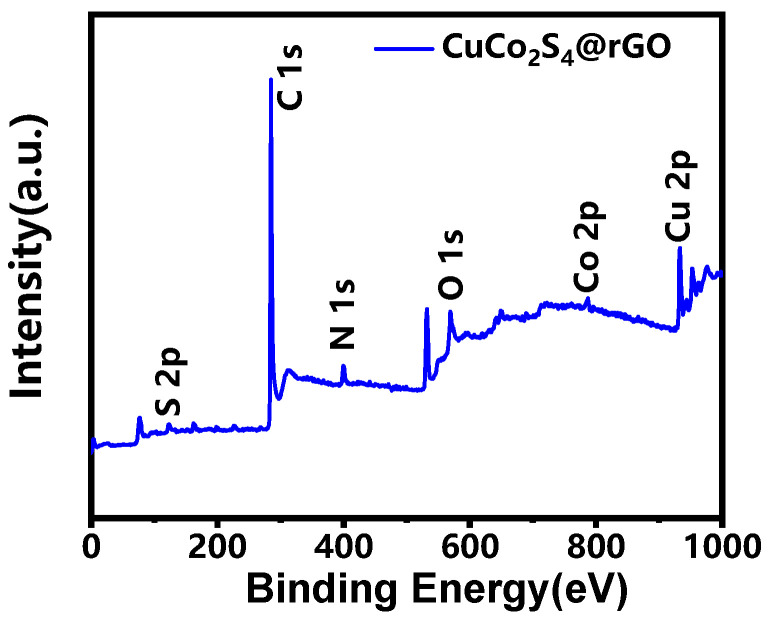
XPS broad scan of CuCo_2_S_4_@rGO samples.

**Figure 6 nanomaterials-14-00182-f006:**
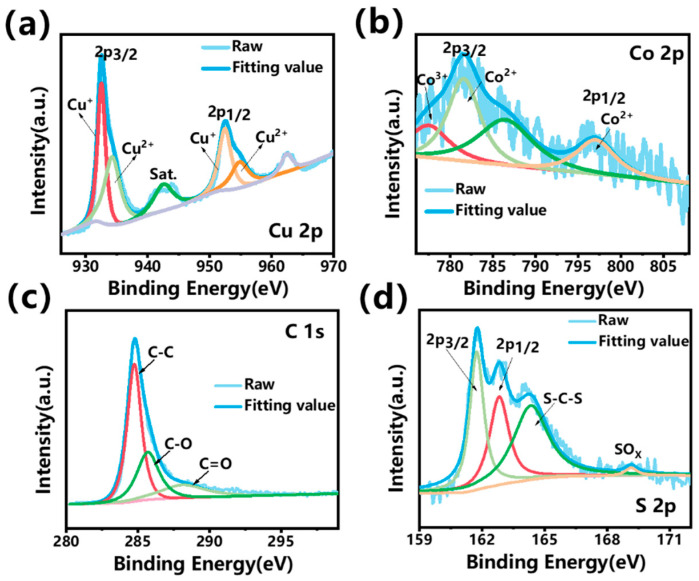
High-resolution XPS spectra of CuCo_2_S_4_@rGO samples: (**a**) Cu 2p; (**b**) Co 2p; (**c**) C 1s; (**d**) S 2p.

**Figure 7 nanomaterials-14-00182-f007:**
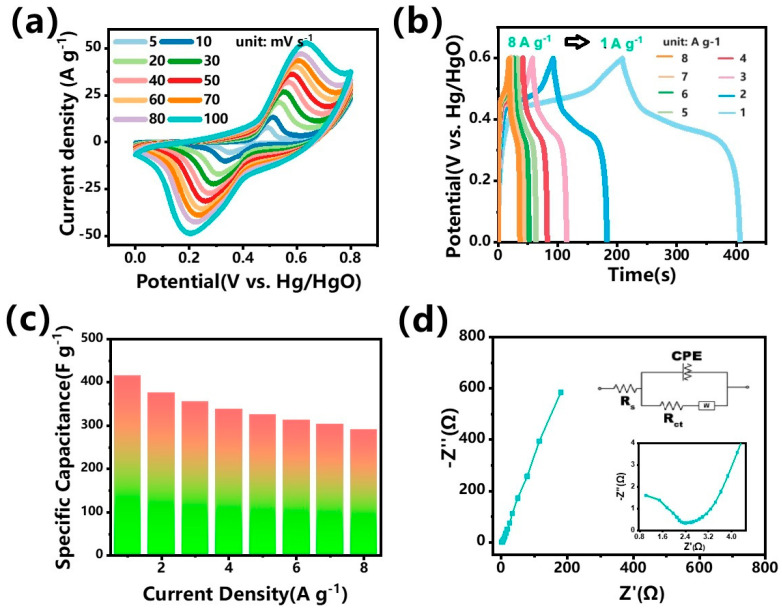
Electrochemical performance of CuCo_2_S_4_@rGO as supercapacitor electrode in the three-electrode cell: (**a**) CV curves of CuCo_2_S_4_@rGO at different scan rate; (**b**) GCD curves of CuCo_2_S_4_@rGO at different current density; (**c**) gravimetric specific capacitance values of CuCo_2_S_4_@rGO based on GCD curves; (**d**) Nyquist plot of the CuCo_2_S_4_@rGO electrode; the inset shows the equivalent circuit.

**Figure 8 nanomaterials-14-00182-f008:**
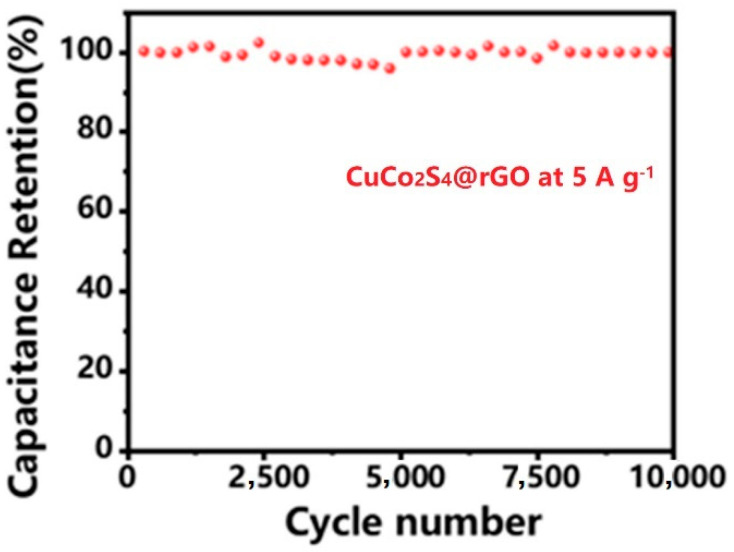
Cycle performance of CuCo_2_S_4_@rGO as supercapacitor electrode in the three-electrode measurement.

**Table 1 nanomaterials-14-00182-t001:** Atomic percentages of the elements in XPS analysis.

Element	Atomic %
C	77.99
O	11.08
N	4.29
Co	1.68
Cu	3.40
S	1.56

**Table 2 nanomaterials-14-00182-t002:** Comparison of the energy storage performance of electrode based on transition metal sulfide composited with graphene.

Electrode	Eletrolyte	Scan Rate/Current Density	Specific Capacitance	Reference
(Ni,Mo)S_2_/G	2 M KOH	1 A g^−1^	376 F g^−1^	[28]
Co_3_S_4_/CoMo_2_S_4_ (CMS)-rGO	6 M KOH	1 A g^−1^	457.8 F g^−1^	[29]
NiS/GO	3 M KOH	1 A g^−1^	400 F g^−1^	[31]
CuCo_2_S_4_@rGO	1 M KOH	1 A g^−1^	415 F g^−1^	This work

## Data Availability

The raw data will be available from the corresponding author upon reasonable request.

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
