# Peer review of "Introducing CuCo2S4 Nanoparticles on Reduced Graphene Oxide for High-Performance Supercapacitor"

_nanomaterials, 2024, doi:10.3390/nano14020182_

Round 1

Reviewer 1 Report

Comments and Suggestions for Authors

The reported spinel copper cobalt sulfides (CuCo2S4) have been widely used as anode materials in both lithium and sodium-ion battery systems because of their considerable storage capacity due to more active sites and richer valence composite. However, pristine spinel has limitations such as poor conductivity and severe volume change during cycling, that restrict the reversible capacity. To address this, the authors have employed a hydrothermal method to prepare CuCo2S4@rGO composite with the aid of the synergistic effect between CuCo2S4 and rGO sheets. The concept itself is not new, this has been reported for battery systems but using it as pseudocapacitive electrode in supercapacitors could be an added insight.

The combination of spinel-structured Cu–Co compounds and graphene to prepare a composite material for both supercapacitors and lithium-ion batteries is a significant area of research. The submitted article has been provided with reasonable physical and electrochemical data drawn from the SEM/TEM morphological images and surface properties like XPS etc. At some parts of the manuscript, it requires some clarification. My concerns are stated below.

·         The rationale for choosing Copper in the spinel oxide must be demonstrated in the introduction.

·         Having a freestanding electrode, can the thickness of the sample be tailored? Given this as the case, then why in section 2.3 PVDF has been used?

·         In Section 1, the reviewed materials like binary transition metal sulfides are excellent, however, a simple approach of binary transition metal oxides (BTMOs) reported as supercapacitor electrodes by Manickam Minakshi et al is also widely researched. Maybe a good idea to include and discuss it widely.

·         What is the role of TAA in the electrochemical reactions?

·         Why to show Figure 3a XRD patterns? It is CuCoO?

·         In Equation 2, what does the letter “a” in CuCo2S4 stand for?

·         Are there any electrochemical data for the electrode system?

·         Figures 7b-c must be explained. In Fig. 7c, what does the color contrast in the bar graph denotes?

·         The pseudocapacitance associated with reversible faradaic reactions and their shape of the CV curve must be referred to the literature reported (such as doi.org/10.1016/j.progsolidstchem.2023.100390; and doi.org/10.1016/j.ceramint.2022.03.266).

·         The Rs and Rct values must be calculated from the EIS plot; and is there any equivalent circuit for the plot shown in Figure 7d?

·         Are the shown SC values in Figure 7c, are proportional to the CD?

·         Page 9, line 278; “room for further optimization” how, please specify.

·         The references are not well formatted.

·         Does the addition of GO in the synthesis process change the morphology?

Comments on the Quality of English Language

At some parts of the manuscript, it can be improved.

Reviewer 2 Report

Comments and Suggestions for Authors

The author's bimetallic sulfide-coupled graphene hybrid, featuring CuCo2S4 nanoparticles on graphene layers with rGO as the substrate, demonstrated outstanding capacitive energy storage. Notably, the CuCo2S4@rGO electrode exhibited a specific capacitance of 410 F g-1 at 1 A g-1 and retained 70% of this capacitance at 8 A g-1, highlighting remarkable rate capability. The electrode's impressive cycling performance, retaining 98% of its initial capacity over 10,000 charge-discharge cycles at 5 A g-1, further validates its efficiency. I recommend the paper for acceptance after minor revision with the following comments:

1-For comparison, stack the XRD graph in one figure and include a stick pattern for CuCo2S4 and GO/rGO in a plot.

2-Which function has been used for XPS fitting?

3-What is the atomic percentage of an element in XPS analysis?

4-Why is the GCD potential window different from the CV profile window?

5-Add an equivalent circuit for the EIS study.

Comments on the Quality of English Language

 Minor editing of English language.

Round 2

Reviewer 1 Report

Comments and Suggestions for Authors

This reviewer reviewed the highlighted parts of the manuscript along with the author's responses to the earlier queries. In this reviewer's opinion, the revised version is suitable for publication.